# The Terahertz Dynamics of an Aqueous Nanoparticle Suspension: An Inelastic X-ray Scattering Study

**DOI:** 10.3390/nano10050860

**Published:** 2020-04-29

**Authors:** Alessio De Francesco, Luisa Scaccia, Ferdinando Formisano, Eleonora Guarini, Ubaldo Bafile, Marco Maccarini, Ahmet Alatas, Yong Q. Cai, Alessandro Cunsolo

**Affiliations:** 1Consiglio Nazionale delle Ricerche, Istituto Officina dei Materiali, Operative Group in Grenoble (OGG), F-38042 Grenoble, France; defrance@ill.eu (A.D.F.); formisano@ill.eu (F.F.); 2Institut Laue-Langevin (ILL), F-38042 Grenoble, France; 3Dipartimento di Economia e Diritto, Università di Macerata, Via Crescimbeni 20, 62100 Macerata, Italy; luisa.scaccia@unimc.it; 4Dipartimento di Fisica e Astronomia, Università di Firenze, via G. Sansone 1, I-50019 Sesto Fiorentino, Italy; guarini@fi.infn.it; 5Consiglio Nazionale delle Ricerche, Istituto di Fisica Applicata ”Nello Carrara”, via Madonna del Piano 10, I-50019 Sesto Fiorentino, Italy; U.Bafile@ifac.cnr.it; 6Université Grenoble-Alpes - Laboratoire TIMC/IMAG UMR CNRS 5525, 38000 Grenoble, France; marco.maccarini@univ-grenoble-alpes.fr; 7Argonne National Laboratory, Advanced Photon Source, P.O. Box 5000 Upton, 11973 NY, USA; alatas@anl.gov; 8Brookhaven National Laboratory-National Synchrotron Light Source-NSLS II, P.O. Box 5000, Upton, 11973 NY, USA; cai@bnl.gov

**Keywords:** inelastic neutron scattering, inelastic X-ray scattering, phonon propagation, nanoparticles, model choice, Bayesian inference

## Abstract

We used the high-resolution Inelastic X-ray Scattering beamline of the Advanced Photon Source at Argonne National Laboratory to measure the terahertz spectrum of pure water and a dilute aqueous suspension of 15 nm diameter spherical Au nanoparticles (Au-NPs). We observe that, despite their sparse volume concentration of about 0.5%, the immersed NPs strongly influence the collective molecular dynamics of the hosting liquid. We investigate this effect through a Bayesian inference analysis of the spectral lineshape, which elucidates how terahertz transport properties of water change upon Au-NP immersion. In particular, we observe a nearly complete disappearance of the longitudinal acoustic mode and a mildly decreased ability to support shear wave propagation.

## 1. Introduction

Although the collective terahertz dynamics of disordered materials has been the focus of thorough investigation in the last five decades, there are still many unexplored areas. For instance, the high-frequency transport behavior of hybrid, liquid–solid, metamaterials is still mostly unknown, despite their promise of novel functionalities. Especially appealing are phononic materials, i.e., materials that can effectively interact with acoustic waves, possibly attenuating, reflecting, or even trapping them. These materials are crucial for the development of a whole new class of thermal devices [1], such as thermal diodes [2], thermoelectrics [3], and thermocrystals [4]. The study of nanostructures exhibiting non-trivial phononic properties is an emerging branch of Condensed Matter Physics, often referred to as terahertz phononics. Being this research field still in its infancy, it is appropriate to consider, as benchmarks for future advances, simple prototypical elementary systems, such as suspensions of small particles in a liquid. It is well-known that, in the macroscopic, or continuous limit, colloids in suspension can efficiently damp the propagation of acoustic waves. Specifically, if the acoustic wavelength largely exceeds the diameter of the colloid, the latter re-radiate density waves like an “antenna”, and the interference of these emissions ultimately causes a nonlinear increase of acoustic damping upon increasing the acoustic frequency. Furthermore, if the elastic constants of the colloid and the hosting medium differ substantially, their acoustic oscillations become mutually dephased, and a viscous drag develops at the colloid interface. The ultimate effect of this phenomenon, customarily referred to as visco-inertial, is to enhance the acoustic damping [5]. Although the influence of floating colloids on acoustic propagation is reasonably understood for lengthscales in a continuum and small colloid diameters [6,7,8,9,10], no comparable knowledge was achieved beyond these approximations. Filling this void seems especially compelling for mesoscopic acoustic waves, which have wavelengths and periods respectively matching first neighboring particle–particle separations and “in cage” rattling periods, and are thus directly coupled with microscopic degrees of freedom of the suspension. These collective excitations are probed by terahertz spectroscopic methods such as Inelastic X-ray (IXS [11]) and Neutron Scattering (INS [12,13]). An IXS or INS measurement can easily zoom in on dynamic events occurring over different scales by proper tuning of the energy and the momentum exchanged between the beam of probe particles (either photons or neutrons) and the sample, i.e., E=ℏω and ℏQ respectively, with *ℏ* being the reduced Planck constant. For low *Q* and ω values, a scattering measurement probes the system’s response as an average over large distances and long time lapses. The probe thus perceives the sample as a continuous and homogeneous medium whose internal constituents have experienced many mutual interactions. In this regime, a hydrodynamic description of density fluctuations [14] holds validity and colloid interconnections are important. For diluted suspensions, these interactions, customarily referred to as hydrodynamic, are primarily mediated by the hosting liquid. Upon further increasing *Q* up to at least $2\pi /d_{c}$, with $d_{c}$ being the colloid’s diameter, the probe gradually focuses on lower wavelength excitations as phonons propagating throughout the colloid interiors.

Owing to this variegated multi-scale dynamics, the sample complexity and the highly damped character of spectral modes of the liquid medium, the interpretation of IXS spectra from colloidal samples is in principle hampered by major uncertainties. Fortunately, Bayesian inferential methods [15,16] can provide an invaluable tool in this endeavor, as recently demonstrated in some spectroscopy neutron and X-ray scattering measurements [17,18,19,20,21]. In one of these works [20], we observed that collective excitations propagating through the pure hosting liquid are severely affected by the presence of even a small amount of nanoparticles in suspension. Specifically, the sample was a dilute aqueous suspension (0.5% volume concentration) of 50 nm-diameter spherical Au nanoparticles (Au-NPs) in water. Its IXS spectrum was analyzed in a relatively limited *Q*-range (3.5 nm${}^{-1}<Q<$ 9.5 nm${}^{-1}$) and compared with literature results on pure water. Being aware of the demonstrated effectiveness of the Bayesian inference in unraveling the dynamics of nanoparticle suspensions, we used this approach to analyze IXS measurements carried out on water and a dilute aqueous suspension of smaller spherical Au-NPs (15 nm diameter). In the present solution, the volume concentration being the same as the one with 50 nm-diameter nanoparticles previously measured, the number of particles in the suspension is much higher, and this might have a visible effect on the overall damping of water excitations.

Indeed, investigating the IXS spectra of the two samples, pure water and nanoparticle suspension, in the same experimental conditions is the ideal prerequisite to single out the effect of immersed Au-NPs on the collective modes of the hosting fluid. Furthermore, the coverage of an expanded *Q*-range (3 nm${}^{-1}\leq Q\leq$ 21 nm${}^{-1}$) enabled these measurements to achieve a more extensive mapping of the complex multi-scale behavior of the samples.

## 2. The IXS Measurement

The IXS measurements were carried out at the HERIX spectrometer at the Sector 30 beamline of the Advanced Photon Source at Argonne National Laboratory. The energy of the incident beam was tuned to ≈23.7 keV, thus matching the energy of the Si (12 12 12) backscattering reflection from both the monochromator and the nine energy analyzers. The analyzers were mounted on the moving extreme of the spectrometer arm, each of them was separated from the adjacent ones by the same angle that corresponded to an ≈2 nm${}^{-1}$
*Q*-offset. The spectrometer arm was positioned so that the analyzers could cover the 3–21 nm${}^{-1}$
*Q* range. The instrumental resolution function was estimated through the spectrum of Plexiglas at the *Q* of its structure factor maximum, where it behaves as an almost elastic scatterer. We found that its scattering signal had a nearly Lorentzian profile with 1.6 meV width (full width at half maximum, FWHM), which was adequate to resolve the inelastic modes of both pure water and suspension spectra even at the lowest *Q*, i.e., 3 nm${}^{-1}$. Further details on the spectrometer can be found elsewhere [22,23]. To prepare the sample, we used 99.9% pure water purchased from Sigma Aldrich, while Nanopartz provided the Au nanoparticles in suspension. To measure the spectra from the suspension with adequate statistical accuracy, energy transfer (*E*) scans typically lasted 6 h. Intuitively, due to the dilute nature of the suspension, about 0.5% in volume NP concentration, one could anticipate that only minimal differences exist between the spectra of aqueous suspension and the one of pure water, also thoroughly investigated in the past by both IXS [24,25,26,27,28,29] and INS [30,31,32,33]. This expectation is clearly at odds with results in Figure 1, showing instead that the two spectral profiles visibly differ when compared after normalization to the intensity maximum. In such a plot scattering profiles measured in pure water (full lines of different colors) and in the Au-NP suspension (dots of corresponding colors) at the same *Q* values are compared to each other with the mentioned normalization.

It can be noticed that the broad side-shoulders of water spectra disappear in the suspension scattering profiles, and are replaced by sharp, yet relatively weak, inelastic peaks. The emergence of these peaks in the suspension spectra is emphasized by the comparison with the spectral wings of pure water, and it seems consistent with the outcome of our recent IXS measurement on an aqueous suspension of Au-NPs mentioned in the Introduction [20]. As discussed therein and further below in this paper, these additional peaks in the suspension spectra directly relate to phonon modes propagating inside the Au-NP interior. At this stage, further considerations on the differences between pure water and suspension spectra should primarily rest on the outcome of a lineshape modelling, which can elucidate, for instance, the different inelastic-to-elastic intensity ratios in the spectra of the two samples and the number of spectral modes contributing to them. In the next two sections, we illustrate how this lineshape analysis was carried out using Bayesian inference methods.

## 3. A Model for the Spectral Shape

In principle, a reliable analysis of the measured spectral shape should primarily be based on a suitable line-shape model, which, unfortunately, for a colloidal suspension is not predicted by any rigorous theory. For this reason, we adopted minimally invasive hypotheses on its shape, assuming that it combines solid-like and liquid-like features, respectively arising from the immersed Au-NPs and the water medium hosting them. Overall, the model shape was a linear combination of elastic, quasielastic, and inelastic contributions. Namely: (1)$$\begin{array}{ccc} S(Q,E) & = & A_{e}\left(Q\right)\delta \left(E\right)+\left[n\left(E\right)+1\right]\frac{E}{k_{B}T}\{L_{A_{0},z_{0}}\left(Q,E\right)+\\ \; & \; & \sum_{j=1}^{k}A_{j}\left(Q\right)DHO_{j}\left(Q,E\right)\}, \end{array}$$
where δ(E) the Dirac delta function of *E* describing the elastic response and having integral $A_{e}$(Q), $n\left(E\right)={\left(e^{E/k_{B}T}-1\right)}^{-1}$ is the Bose statistics factor expressing the detailed balance condition; finally, the term between curly brackets is the sum of a Lorentzian quasielastic contribution—having half width at half maximum $z_{0}$ and area $A_{0}$ and *k* inelastic contributions, accounted for by Damped Harmonic Oscillator ($DHO_{j}\left(Q,E\right)$) terms having areas $A_{j}\left(Q\right)$:(2)$$\begin{array}{c} DHO_{j}\left(Q,E\right)=\frac{2}{\pi }\frac{\Omega_{j}^{2}\left(Q\right)\Gamma_{j}\left(Q\right)}{{\left(E^{2}-\Omega_{j}^{2}\left(Q\right)\right)}^{2}+4{\left[E\Gamma_{j}\left(Q\right)\right]}^{2}}, \end{array}$$
where $\Omega_{j}\left(Q\right)$ and $\Gamma_{j}\left(Q\right)$ are the undamped energies and the damping coefficients of the *j*th DHO excitation. Notice that the number *k* of the $DHO_{j}\left(Q,E\right)$ excitations appearing in the model and their shape coefficients are equally treated as adjustable parameters.

We recall here that, by definition, a DHO profile has no separate central peak, which must be added to it to reproduce the extended hydrodynamic triplet observed in the typical terahertz spectrum of a disordered system. Here, such a central peak is modelled by the sum of an elastic and a quasielastic peak, both options being, in principle, poorly justifiable approximations for pure water. Although the Lorentzian profile can reproduce consistently the merely relaxational Mountain mode [34], its combination with the DHO terms yields a spectral shape with a diverging second spectral moment. For this reason, such a model should fail to reproduce the extreme spectral tails, which extend beyond the relatively narrow energy range covered by our line-shape best-fitting analysis (−20 meV ≤ E ≤ 20 meV). In addition, a ∝δ(E) term in the spectrum of water is hardly justifiable on a rigorous basis in the (Q,E) values and sample thermodynamic conditions explored. Nonetheless, it was still included in the model for two main reasons: (1) we decided to use, for pure water, a model option formally consistent with the one used for the suspension spectrum, which has a non-negligible elastic component arising from the water-Au interfacial scattering. (2) Most importantly, we adopted an unbiased approach, ultimately entrusting the fitting outcome with the final assessment on the probabilistically most grounded model option. Of course, the latter could still be a posteriori critically evaluated against available literature results.

To describe accurately the measured spectrum, the model function in Equation (Equation 1) must be convoluted with the instrument resolution function R(Q,E) and a spectral background should also be added to such a convolution. Explicitly:(3)$$\tilde{S}\left(Q,E\right)=R\left(Q,E\right)⊗S\left(Q,E\right)+B\left(E\right),$$
where B(E) is a typically mildly *E*-dependent background intensity. In the following section, we will illustrate how a Bayesian inferential procedure can be applied to determine, based on the measurements and on our prior knowledge of the physical problem under investigation, the most plausible shape, intensity, and background parameters entering in Equation (Equation 3) through Equations (Equation 1) and (Equation 2).

Best-fitting model lineshapes were determined using a Bayesian inferential analysis implemented through a Markov chain Monte Carlo (MCMC) routine with reversible jump (RJ) steps, described in some detail in Ref. [35]. This approach can be used to probabilistically infer, based on a given measurement, the joint posterior probability distribution, or, in short, the posterior, of each model parameters. Notice that among model parameters we here included the number of excitations (*k*) contributing to the spectrum. In this respect, the Bayesian procedure can be used to probabilistically “rate“ a guess on the number of excitations in the sample’s scattering signal. More in general, the knowledge of the entire posterior distribution of each model parameter also authorizes interpreting the maximum of such a posterior, as the optimal (most plausible, or best-fitting) value of such a parameter. This assignment is possible provided the posterior, albeit not necessarily symmetric, is sharply peaked, well-shaped, and unimodal. The shape itself of the posterior carries important information on the precision and the likelihood of a given best-fitting value. Foundations and working principle of the performed Bayesian analysis are discussed in great detail in Reference [19].

## 4. Discussion of Results

In Figure 2, the IXS spectral shapes measured either on the Au-NP suspensions (left column) or in pure water (right column) are compared with corresponding best-fit lineshapes along with their spectral components, as defined in Equation (Equation 1). As discussed, these best-fit lineshapes refer to the model option most often visited by the algorithm, which in the present case corresponds to *k* equal to either 1 or 2, i.e., containing one or two DHO terms, respectively. The broad high-energy shoulder in the pure water is customarily assigned to the longitudinal acoustic (LA) mode, and its phenomenology was thoroughly investigated by IXS [24,25,29]. Conversely, high-energy peaks in the suspension spectra are substantially narrower than their counterparts in typical liquid systems, and we thus assign them to Au phonon modes propagating inside the NPs’ interior [20]. As discussed further below, this assignment also is based on the similarities between the *Q*-dispersion curves of this mode and the one of liquid Au. Of course, the weak intensity of Au phonons in the spectra mirrors the sparse concentration of Au-NPs in the considered suspension.

### 4.1. The Low-Energy DHO (Cyan Lines in the Plots)

For both pure water and colloidal suspension, this excitation arises from the coupling of density fluctuations with shear or transverse waves [26,36]. From Figure 2, it can be noticed, consistently with previous findings [26], that this spectral feature emerges at sufficiently large *Q* and eventually becomes the dominant inelastic feature of both pure water and colloidal suspension spectra. Overall data in Figure 2 suggest that: (1) for the Au-NP suspension, the onset of this spectral feature occurs at *Q* values slightly higher than in pure water, and (2) its damping seems higher at some *Q*’s. These facts suggest that dispersed Au-NPs might also have a slight damping effect on the transverse acoustic propagation in water.

Concerning the Lorentzian quasielastic term (wine line), it appears that in most cases it dominates the spectral shape of both samples; it was proven in the literature that this feature arises from the coupling of density fluctuation with a structural relaxation active in water [24,26,28,29]. For water at ambient conditions, this relaxational mode dominates the quasielastic part of the IXS intensity [28]. However, a weaker Lorentzian contribution might be detectable, and it represents the finite-*Q* extension of the Rayleigh hydrodynamic mode, relating to thermal diffusions triggered by spontaneous temperature gradients in the liquid.

Best-fit values of the DHO frequencies $\Omega_{1,2}$ are plotted in Figure 3 as a function of *Q* as obtained for pure water and for the Au-NP suspension. To avoid ambiguities, we will quote $\Omega_{1}$ and $\Omega_{2}$ as the low and high energy inelastic shift, respectively. The interpretation of the *Q* dependence of these shape parameters is straightforward and directly follows from the comparison with their counterparts in heavy water [37] and liquid gold [38], also reported in the plot for reference. It can be readily appreciated that, for Au-NP suspension, *Q*-dependent values of $\Omega_{1}$ are very close to the transverse acoustic mode (TA) dispersion of pure water, while the one of $\Omega_{2}$ closely resembles the sound dispersion of liquid Au. We thus conclude that:(1)The resemblance between $\Omega_{1}$ values of the Au-NP suspension with the transverse acoustic mode of water further endorses the assignment of this mode to shear waves propagating through the liquid medium of the aqueous suspension.(2)Conversely, the similarity of $\Omega_{2}$ with the dispersive branch of liquid Au further endorses the assignment of the high-energy peaks in the Au-NP suspension spectrum to acoustic phonons propagating inside the Au-NP interiors. One might find somehow baffling the similarity of the phonon propagation in liquid Au with the one in its solid counterparts, i.e., in the Au-NP interiors. However, this might be consistent with a viscoelastic hypothesis, assimilating the high-frequency response of a liquid to the one of the corresponding solid, as previously observed in water [39].

It can be readily noticed that the dispersion curves of water and liquid gold exhibit striking differences, as, for instance, markedly different apical values, ≈18 meV and ≈28 meV, respectively. This difference can be explained by the substantially dissimilar Einstein frequency. This property characterizes the dispersive behavior of condensed matter systems, and, in a first approximation, gives an estimate of the maximum frequency (energy) of the dispersion curve [40]. Indeed, it is especially high in fluids having a harsh repulsive portion of the interatomic potential, as water and noble gases, while it is substantially smaller in liquid metals, which have a relatively soft core repulsion [41].

Figure 3 includes for reference the two linear hydrodynamic dispersions $c_{s}Q$ of liquid and solid Au, as obtained using the values of the adiabatic sound speed $c_{s}$ in Refs. [42,43], respectively. The circumstance that at low and moderate *Q*’s the dispersion curve of liquid gold is steeper than the corresponding hydrodynamic dispersion and rather similar to the solid Au one mirrors its viscoelastic behavior; this manifests itself through a *Q*-transition of the dispersion curve from a liquid-like (viscous) regime to a solid-like (elastic) one, which is the one primarily probed by current measurements. At higher Q’s, the persisting proximity of $\Omega_{2}$ values in the suspension with the dispersion curve of liquid Au suggests to extend its identification with the frequency of phonons in the NPs.

### 4.2. The Onset of the Transverse Mode in Pure Water

Concerning this point, the outcomes of the present work seem to challenge literature results in two main aspects. First, they show that IXS can be successfully used to characterize TA propagation in water even at ambient conditions. Conversely, previous MD [36] and IXS [25,26] studies suggested that shear propagation modes in the spectrum of water become visible only at lower temperatures, likely due to: (1) the higher hydrogen bond network rigidity and its enhanced ability to propagate shear stresses, and (2) the narrower spectral wings of the quasielastic relaxation and thermal diffusion modes which improve the visibility of low-energy inelastic features.

Second, the present results demonstrate that, in water at ambient temperature, the coupling of density fluctuation with a shear dynamics persists down to *Q*’s slightly lower than previously suggested by a joint IXS-INS measurement [37], which located the *Q*-cutoff for this coupling at 6 nm${}^{-1}$, as shown in Figure 3. Conversely, the outcome of the present Bayesian analysis suggests shifting such a cutoff down to a value as low as 3 nm${}^{-1}$. Indeed, Figure 4a–c summarize the result of the Bayesian inference obtained for the *Q* = 3 nm${}^{-1}$ spectrum of pure water, for which the model option most visited by the algorithm contained three DHO profiles (i.e., it corresponded to *k* = 3). As clear from Panel a, two out of the three DHO’s, DHO ${}_{2}$ and DHO ${}_{3}$, have the form of sharp underdamped doublets, while the lowest DHO mode, DHO ${}_{1}$, appears instead to be single-peaked upon resolution convolution. As mentioned, within the proposed Bayesian approach, the dominant energies of these excitations, $\Omega_{1}$, $\Omega_{2}$, and $\Omega_{3}$, can be derived as the maxima of the corresponding posterior distributions, which occurs at 0.89 ± 0.05 meV, 4.3 ± 0.2 meV and 6.2 ± 0.2 meV, respectively (see Figure 4b). The second value nicely matches the high-energy dispersive branch of water in Figure 3 and assigned to the LA mode. At this stage, it seems unclear why, in the privileged model option, the LA splits into two narrow contributions, one of which sits at an energy ≈ 4 meV. It is not unreasonable that this is an artifact due to the limited statistical accuracy in the spectral counting. Finally, the lowest energy DHO, $\Omega_{1}$, has a dominant energy compatible with a low-*Q* extrapolation of the TA branch of water. Nevertheless, it clearly appears that the proper discernment of the DHO ${}_{1}$ profile in Figure 5a is severely hampered by resolution-limitations. In fact, at this low *Q* value, the inelastic DHO ${}_{1}$ doublet yields a contribution whose FWHM is barely larger than the resolution one, and, upon convolution with the resolution, it transforms into a single featureless peak.

It is noteworthy that the posteriors in Panel b are unimodal and remarkably sharp, thus supporting the high plausibility of the optimal model option displayed in Panel a. Finally, Panel c illustrates the variation of the parameter value upon increasing the number of algorithm’s sweeps (traceplots). The flat trend followed by the curve pinpoints the proper convergence of the algorithm to the privileged model option and parameters estimate.

### 4.3. The Inherent “Fragility” of the LA Mode of Water

One of the most relevant results discussed thus far is that the LA mode of water is seemingly more directly affected by the presence of Au-NPs, to the extent that it nearly disappears upon immersion of an even small amount of them. In a recent IXS work [21], we also demonstrated that the LA mode disappears upon confinement of water in carbon nanotubes, whereas the TA mode persists under this highly directional confinement. Regardless of the presence of a nanostructure, as shown in Figure 2, the LA mode of water rapidly gets more damped and weak upon increasing *Q* up to completely disappear at high *Q*’s where the transverse mode becomes largely dominant. This trend is not entirely surprising as one considers that longitudinal and transverse modes of water respectively couple with the hydrogen bond stretching and bending, or hindered rotations. It is reasonable to expect that, upon increasing *Q*, the contribution of intermolecular rotations gradually dominates over molecular center of mass translations.

Given the results discussed thus far, the expected disappearance of the TA mode (DHO${}_{1}$) of water at Q→0 might follow at least two different pathways: (1) its area might gradually vanish over long distances, or (2) it might remain finite, but the mode could become overdamped, reflecting the inability of the system to support shear propagation over large distances.

Data in Panel a of Figure 5 seem to support the second hypothesis, as they suggest that, for pure water (red dots), the area of the DHO${}_{1}$ term ($A_{1}$) normalized to the one the whole spectrum (*A*${}_{tot}$) not only remains finite, but also substantially increases at the lowest *Q*’s. However, the mentioned resolution-limitations hamper a firm definition of the DHO${}_{1}$ shape at the lowest *Q*’s, which does not authorize to exclude a priori that the damping of the TA mode is critical or sub-critical ($\Omega_{1}\leq \Gamma_{1}$) already at 3 and 5 nm${}^{-1}$ (open circles). This being the case, such a spectral feature becomes hardly distinguishable from the dominating Lorentzian relaxation mode [37]. Indeed, for the lowest *Q* spectra, resolution-limitations might have misled the Bayesian inference analysis.

Interestingly, the low *Q*-rise of $A_{1}/A_{tot}$ can only be appreciated for pure water data, while, for their suspension counterparts (black dots), the presence of some maximum at *Q* = 11 nm${}^{-1}$ can be inferred instead. This difference suggests that immersed Au-NPs might have a sizeable influence on shear wave propagation, as also indicated by the slightly higher *Q* value at which the transverse mode emerges in the spectrum of the suspension. Best-fit values of the halfwidth $\Gamma_{1}$ are reported in Panel b of Figure 5. Despite important data oscillations, it can be noticed that, for both samples, this parameter decreases upon lowering *Q*’s, even though such a trend seems regular and monotonic for pure water only. Relatively scattered values of this parameter are not unexpected, as the typically loose inelastic features of IXS measurements often challenge a precise determination of the inelastic halfwidths. Best fitting parameters reported in Figure 5c,d are respectively the same as in Figure 5a,b, but they refer to the high-energy excitation, DHO${}_{2}$. Notice that the DHO${}_{2}$ excitation has a different assignment for the pure water and the suspension, being connected, in the respective cases, either with longitudinal acoustic waves in water, or with longitudinal phonons propagating in the Au-NP interior. This is certainly true at sufficiently high *Q*’s; in fact, our assignment stems from the mere inspection of shape parameters, which turns out to be markedly different in the two samples only for Q> 5 nm${}^{-1}$. The circumstance in which the weak Au phonon peaks (see Figure 1) dominate the high-energy wings of the suspension spectrum highlights the parallel attenuation of the longitudinal mode of water caused by Au-NPs in immersion. Finally, it appears that, in pure water, the longitudinal mode gradually loses visibility upon increasing *Q*, as deduced from the decrement of *A*
${}_{2}/A_{tot}$ (Figure 5c) and the parallel enhancement of $\Gamma_{2}$ (Figure 5d).

It is noteworthy that our previous IXS measurement on a dilute aqueous suspension of larger (50 nm diameter) Au-NPs [20] evidenced the presence of a sharp low energy feature assigned to the TA phonon peak of gold. The absence of this mode in the currently measured lineshapes could be ascribed to the imperfect crystallinity of the smaller Au-NPs considered here. Indeed, it is known [44] that atoms of Au-NPs arrange themselves in an ordered fcc-type structure resembling that of bulk gold, yet with an appreciable local disorder which increases upon decreasing the NP size, also enhancing in the presence of water. Current results urge us to conclude that this lower structural coherence might be mainly responsible for the suppression, or the considerable reduction, of the TA peak in the spectrum presently measured.

## 5. Conclusions

In conclusion, we presented here the results of an Inelastic X-ray Scattering (IXS) measurement of the spectrum of density fluctuation of pure water and a dilute suspension of 15 nm diameter Au-NPs in water. The comparison of the scattering signals from the two samples demonstrates that, despite the low concentration of about 0.5% in volume, immersed Au-NPs substantially change the spectral shape of water, whose inelastic portion is dominated by two acoustic-like modes having either a longitudinal or a transverse polarization. While the former excitation nearly disappears upon Au-NP immersion, the latter seems to be more mildly affected. Effects of immersed Au-NPs on the transverse acoustic mode in the water spectrum are possibly the shift of its onset to larger *Q* values and an increased damping. Parallel to changes on the inelastic modes of water, the presence of Au-NPs also gives rise to sharp, yet relatively weak, inelastic peaks showing a dispersion that we have compared with the one of solid Au and the viscoelastic dispersion of liquid Au. In this way, we could ascribe these excitations to longitudinal Au phonon modes propagating through the Au-NP interior. The absence of its transverse counterparts in the measured spectra likely owes to finite-size lattice distortions in the Au-NPs considered.

On a more general note, regardless of the presence of Au-NPs, our results indicate that shear wave propagation provides the overwhelming contribution to acoustic propagation at sufficiently high *Q* values. Here, LA waves become strongly damped and weak to merge into the spectral background. As a natural extension of this work, one can envisage systematic investigation of these lineshape effects by changing features of the suspension such as the hosting liquid, or the nanoparticles’ size, shape, and concentration.

## Figures and Tables

**Figure 1 nanomaterials-10-00860-f001:**
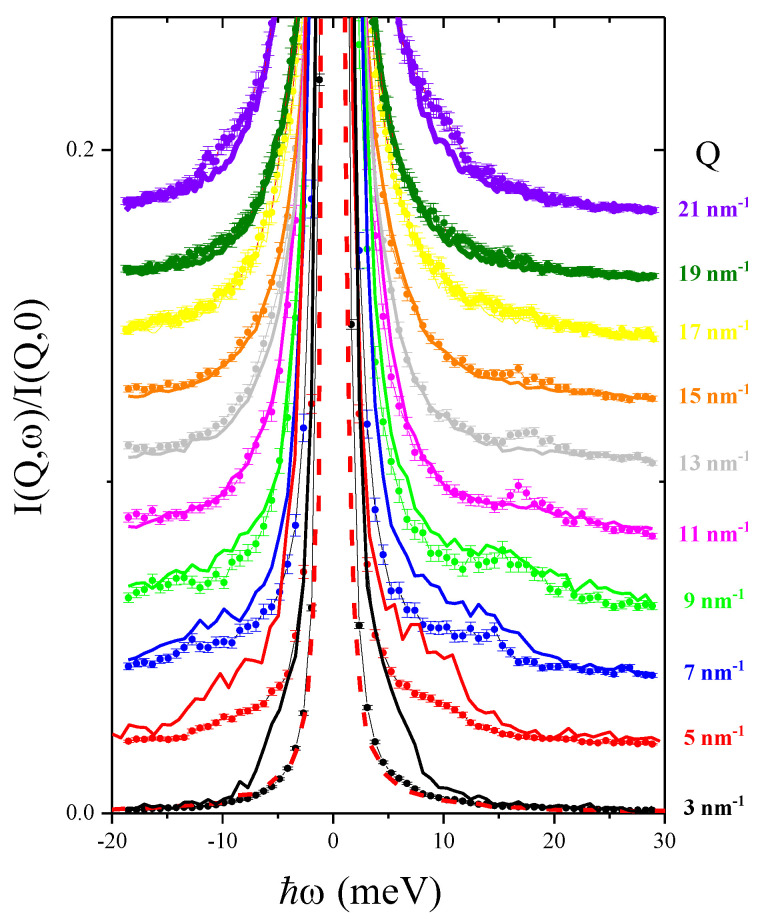
A cascade of spectra measured in the aqueous suspension 15 nm diameter Au-NPs (dots and error bars) and pure water (solid line) at the *Q* values specified in the plot. Data are normalized to the respective maxima and vertically offset by an amount proportional to *Q*. The lowest *Q* spectra are also compared with the energy resolution function after a similar normalization (red dashed curve).

**Figure 2 nanomaterials-10-00860-f002:**
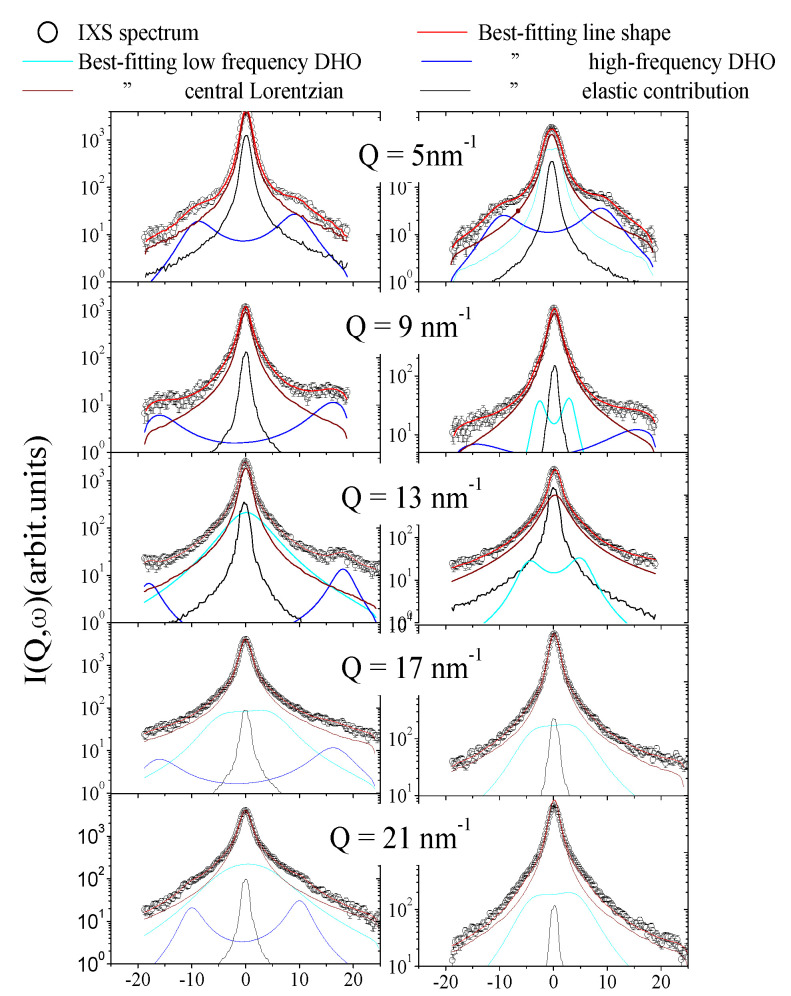
The IXS spectra of the Au-NP suspension (**left column**) and the pure water (**right column**) are compared with best fit lineshapes and the individual spectral components defined in Equation (Equation 1), as indicated in the legend.

**Figure 3 nanomaterials-10-00860-f003:**
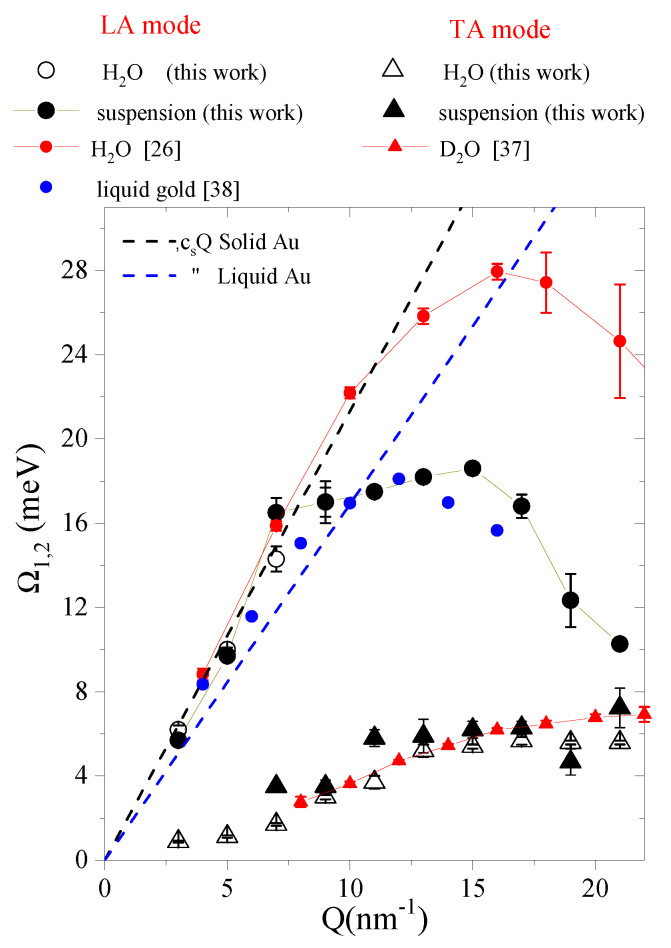
The dispersion curves derived through the best fit values of the DHO frequencies $\Omega_{1,2}$ for the Au-NP suspension and pure water are compared with the low and high energy dispersion branches of heavy water [37], customarily ascribed to a transverse acoustic (TA) and longitudinal acoustic (LA) modes, as well as with the (LA) branch of liquid gold [38]. The meaning of the various symbols are specified in the legend.

**Figure 4 nanomaterials-10-00860-f004:**
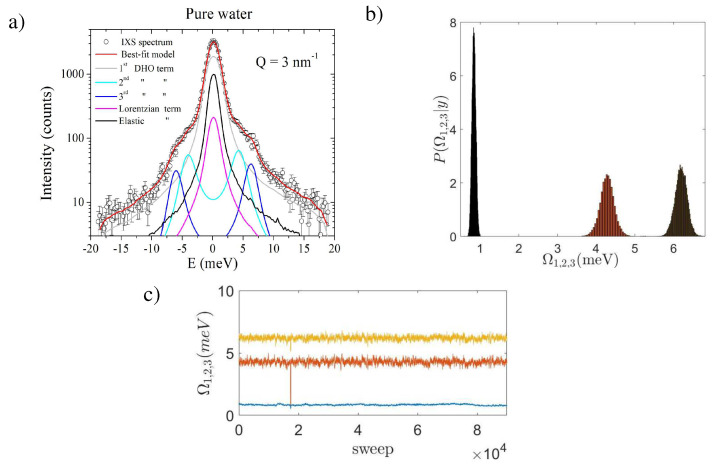
(**a**) the IXS spectrum of pure water measured at *Q* = 3 nm${}^{-1}$ is compared with the best-fit lineshape and its individual components (see Equation (Equation 1)); (**b**) posterior distributions associated with the dominant energies $\Omega_{1}$, $\Omega_{2}$ and $\Omega_{3}$ of the three DHO profiles; (**c**) the variation of the parameter value as a function of the algorithm’s sweeps.

**Figure 5 nanomaterials-10-00860-f005:**
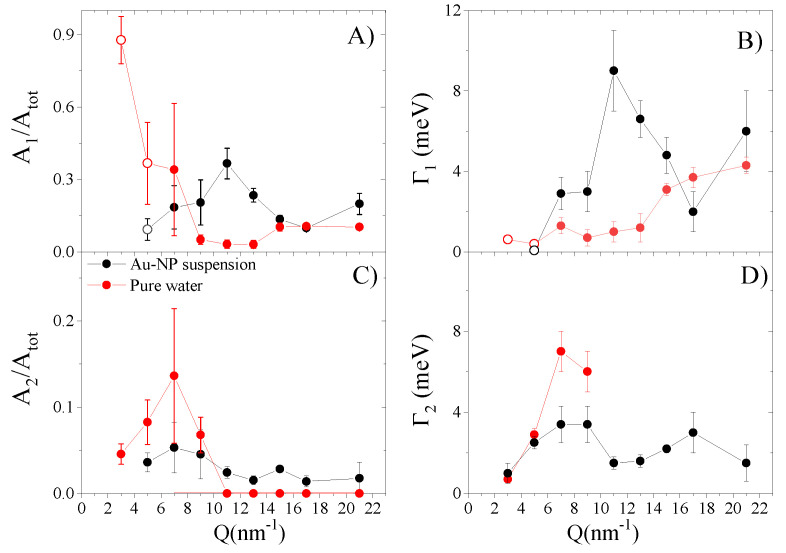
(**A**) displays, as a function of *Q*, the best-fit areas, A${}_{1}$ of the low-energy inelastic profile, DHO${}_{1}$, normalized to the one of the whole spectrum, $A_{tot}$; (**B**) shows the halfwidth of the inelastic peaks, $\Gamma_{1}$ of the same DHO${}_{1}$ excitation. Values reported in (**C**,**D**) are as those in (**A**,**B**), respectively, but in reference to the high-energy inelastic profile DHO${}_{2}$. The open circles label resolution-limited best fit values, as discussed in the main text.

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
