# Peer review of "The Terahertz Dynamics of an Aqueous Nanoparticle Suspension: An Inelastic X-ray Scattering Study"

_nanomaterials, 2020, doi:10.3390/nano10050860_

Round 1

Reviewer 1 Report

To say it in advance, the scientific background of this manuscript is interesting and the scientific topic is well presented. The experimental procedure and data reduction are also carried out professionally, as expected from this experienced team. In general, therefore, the manuscript is definitely worth publishing. Also, the approach to use the Bayes theorem in the attempt to obtain optimal model fitting in the analysis of inelastic scattering data is very advantageous, as has been shown in some recent publications. However, I am struggling with the presentation of the theoretical framework for the data analysis in Chapter IV. I doubt that the average reader of the journal will be able to follow the theoretical statements given there. This part of the manuscript is not well written. The sentences are often much too long and often appear confusing at first glance. Terms appear which have not been explained before and different terms are used for one and the same parameter, which is also confusing. Just as one example: I managed to follow the explanations up to eq. (5). But then I could no longer understand the paragraph underneath at all. I could only guess what it might mean. Let me try to translate this text into my vocabulary, and please tell me if this was originally meant that way:

“In fact, any available a priori information about the problem, embodied by the vector I, is to be used in the Bayesian theorem. However, if such information is not available, uninformative priors (uniform) can be used instead. In the present case, however, only the results of a conventional least squares fit would then enter into eq. (4). Nonetheless, is a priori information still included into the procedure via the second r.h.s. term of eq. (4), which should not be underestimated.”

I am not sure if this is what the authors wanted to express. But even then, I am still struggling with the terms “uninformative priors (uniform)” which have not been defined in the previous paragraphs. It is such passages that make the text very difficult to understand.

The following text then reports on the solution of the Bayesian equation which is of course not solvable analytically. Hence, the authors seek for a numerical solution which they find in using a Markov-Chain Monte-Carlo approach. This procedure has already been proven successful in earlier work. This passage is also difficult to read. It contains information from other, earlier publications, which was much better explained there. However here, the contextual connection is extensively missing, and only comprehendible if the preceding work has been consumed before. 

The authors should understand that not every reader is an expert in this type of Bayesian data analysis, but might still be interested in a general understanding of the procedure. Less detail and rather a clear understanding of the data analysis would be helpful. Readers who are interested in details could be referred to previous work. However, the present text is neither one nor the other. Another option, but less desirable from my point of view would be, to fully dispense the chapter and only refer to corresponding, already published work.

The discussion of the results in chapter V is quite straight forward. The DHOs reveal two kind of modes in pure water as well as in the suspension. In pure water these are interpreted as a low frequency TA-excitation and a high frequency LA-excitation, respectively. The TA-dispersion is in good agreement with literature data but extends to lower Q-values which is a new and interesting observation. However, I am missing the LA-value of pure water at 9 nm-1. The maximum is visible in the right column of Fig. 2 but the frequency is not shown in Figure 3. It might be hidden behind a black dot of the Gold-NP LA-dispersion. But then it does not follow the expected literature values for the pure water LA (red dots). Perhaps I am somehow wrong, but the authors should explain this.

In the suspension, there are also two excitations. Here, the low energy excitation is identified as the aforementioned water-TA. The high energy excitations however, appears to be very similar to what has been interpreted before as an LA-dispersion in liquid Gold. Hence, the authors interpret this feature as resulting from collective acoustic excitations inside the gold nano particles, which is also a new finding.

According to the remarks above, the manuscript should be thoroughly revised before publication and the authors should answer the questions asked.

Reviewer 2 Report

This paper deals with the terahertz dynamics of Au nanoparticle suspension. High-resolution inelastic X-ray scattering spectra of the suspension were analyzed by the Bayes inference method and the collective molecular dynamics of water in the suspension was compared with that of pure water. The effect of Au nanoparticle on the longitudinal and transverse acoustic modes in the suspension were elucidated. The authors have previously measured a suspension of 50 nm diameter Au nanoparticle and also discussed the size dependence of the phonon propagation of nanoparticle suspension. These outcomes are new and give basic information for the development of phononic materials.

I have some comments to improve the paper.

I propose that the Q dependence of the strength and the damping coefficient of each DHO component obtained from eqation2 are shown. They are interesting for the researchers of the molecular dynamics of liquids. The authors concluded that the disappearance of the longitudinal acoustic mode and change of transverse counterpart by addition of Au nanoparticles. The strength and the damping coefficient of DHO component would be necessary to support the conclusion.

As a minor point, please show the horizontal axis of Fig, 4 b.

Reviewer 3 Report

This paper is fairly well written. However, there are a few issues with the presentation of physics. In Fig. 3, the data of "this work" should be more pronounced and not hidden behind the reference data. Also in the annotation at the top, it is better to have the data of this paper (this work) at the top and the reference data further down. Furthermore, the differences between the dispersion curves of Au and H2O should be better discussed. Why do the curves have apex at different energies, ~18 meV and ~28 meV and why do they decrease at high Q and not the dashed lines (models?). Do the authors expect that their work will have an apex at the same energy as in ref. 26 and why is not new high Q data presented? The TA branch should also be mentioned in the legend.  

Round 2

Reviewer 3 Report

The authors have followed the advice of the reviewer and extended the manuscript with new information. Some of the new information deserves to be mentioned in the conclusions.

Author Response

Dear Editor,

We thank again all referees for their helpful remarks, which gave us the possibility to make clearer the content of our study.

We submit a slightly revised version of the manuscript in which the main changes made upon Referee 3 suggestions are mentioned in the Conclusion Section. Specifically, a sentence therein has been rephrased as follows (changed/added part in red fonts):

"In parallel to changes on the inelastic modes of water, the presence of Au-NPs also gives rise to sharp, yet relatively weak, inelastic peaks showing a dispersion that we have compared with the one of solid Au and the viscoelastic dispersion of liquid Au. In this way, we could ascribe these excitations to longitudinal Au phonon modes propagating through the Au-NPs interior."

 Sincerely,

Alessandro Cunsolo